# Using Multi-Factor Analysis to Predict Urban Flood Depth Based on Naive Bayes

**Huiliang Wang, Hongfa Wang, Zening Wu and Yihong Zhou ***

College of Water Conservancy Engineering, Zhengzhou University, Zhengzhou 450001, China; wanghuiliang@zzu.edu.cn (H.W.); w18557519093@163.com (H.W.); zeningwu@zzu.edu.cn (Z.W.)
* Correspondence: zhouyhong0214@163.com

**Abstract:** With global warming, the number of extreme weather events will increase. This scenario, combined with accelerating urbanization, increases the likelihood of urban flooding. Therefore, it is necessary to predict the characteristics of flooded areas caused by rainstorms, especially the flood depth. We applied the Naive Bayes theory to construct a model (NB model) to predict urban flood depth here in Zhengzhou. The model used 11 factors that affect the extent of flooding—rainfall, duration of rainfall, peak rainfall, the proportion of roads, woodlands, grasslands, water bodies and building, permeability, catchment area, and slope. The forecast depth of flooding from the NB model under different rainfall conditions was used to draw an urban inundation map by ArcGIS software. The results show that the probability and degree of urban flooding in Zhengzhou increases significantly after a return period of once every two years, and the flooded areas mainly occurred in older urban areas. The average root mean square error of prediction results was 0.062, which verifies the applicability and validity of our model in the depth prediction of urban floods. Our findings suggest the NB model as a feasible approach to predict urban flood depth.

**Keywords:** causes of the flood; different return periods; multiple factors analysis; Naïve Bayes; urban flooded prediction





## 1. Introduction

The acceleration of urbanization reshapes cities and brings new opportunities for the development of the economy and society. At the same time, urbanization gradually increases the urban waterlogging problem. This problem becomes even more alarming because of climate change. For example, urban floods around the world, such as the New York floods in 2011, the Houston floods in 2017, the Rio floods in Brazil and the South Sulawesi floods in Indonesia in 2019, were caused by a combination of the greenhouse effect, urbanization, land-use change, and other factors [1,2]. When floods occur in cities with high economic and population density, they cause serious property losses, disruption of the production of goods and services, casualties or loss of life, and the hindering of urban development [3]. As the largest developing country in the world, China also frequently experiences natural disasters and is more affected by flooding than any other country [4], e.g., the Chinese Sichuan and Jiangxi floods in 2020. In 2018, 31 Chinese provinces (autonomous regions and municipalities directly under the Central Government) experienced flood events, affecting 55.77 million people and flooding or waterlogging 83 cities, resulting in a direct economic loss of 23.1 billion dollars (a value that corresponds to 0.18% of the annual Chinese GDP) [5]. According to the fifth IPCC (Intergovernmental Panel on Climate Change) climate assessment report, climate change will increase the occurrence of extreme weather events during the 21st century [6], one of which is urban floods. Therefore, predicting the characteristics of a rainstorm plays a crucial role in identifying possible risks and thereby warning the city's residents to avoid or reduce activity in areas at a high risk of waterlogging.

Many studies on flood prediction based on deterministic numerical models have made considerable progress in reproducing flood situations [7]. In general, the most common methods used to simulate and predict the spatial distribution of floods is to use hydrodynamic and hydrological models [8]. Hydrologic models usually analyze hydrological features and describe runoff confluence physically. Based on the hydrodynamics theory, researchers usually derive confluence equations by combining the physical laws of mass, momentum, and energy conservation [9]. Representatives of these models are the Storm Water Management Model (SWMM) [10], MIKE SHE [11], and Urban Flood Cell Model (MODCEL) [12]. However, these models demanded the determination of several parameters to achieve adequate accuracy levels [13], thus limiting their application. Hence, other simple and effective methods are gradually being developed and used by people.

With the development and maturity of computer technology, people begin to develop data-driven intelligence models to successfully predict hazards and risks in environmental sciences [14]. Data-driven intelligent models mainly analyze existing observation data to build input–output mapping relations and predict specific hydrological quantities. Hence, various data-driven intelligence models, such as artificial neural networks such as the Elman neural networks [15,16] and the nonlinear autoregressive network with exogenous inputs (NARX) network [17], support vector machines (SVM) [18] and deep learning involving the Random Forest [19,20] and gradient-boosting decision tree (GBDT) [21] have been widely applied for urban flood risk assessment and prediction in recent years with satisfactory results. Urban flood prediction mainly includes three flood characteristics—the duration of the flood, the area affected by flooding, and the depth of the flood [22]. In actual urban flood management, we can actively respond to emergencies and greatly reduce the negative impact of flooding if we can determine the maximum depth of urban flooding during a rainfall event. A variety of methods can be used to predict urban flood depth, but complex structures and effective predictions require time and computational resources. Even small processes can put massive demands on the computer [23]. Naive Bayes (NB) has shown exceptional speed and accuracy with large data sets [24]. However, few studies have used NB to predict the depth of urban flooding.

With this in mind, we used it as an effective tool for the study of urban flood depth prediction. NB is an algorithm based on Bayes' theorem to find the optimal result under the maximum probability. It transforms the multi-dimensional condition into a multiplicative relationship of multiple conditional probability distributions. The main advantages of using an NB model is its calculation speed; its robust, minimal number of parameters; and that it is easy to understand [25,26].

In the current paper, we applied NB to urban flood depth prediction. To evaluate model performance, we used the pass rate (QR), Nash efficiency coefficient (NSE), and root mean square error (RMSE) of the predicted and measured values to evaluate NB model performance in validation sets. Compared with the traditional hydrological model SWMM, the NB model showed better performance. Finally, the trained NB model was used to predict the flood depth of designed rainfall under different rainfall recurrence periods. The findings presented here can provide effective technical support for the control and management of urban floods. Under changing urban drainage systems, only physically based models would be successful. Thus, this research and the analysis has been carried out under the background that the urban drainage system has not changed.

## 2. Materials and Methods

### 2.1. Study Area

Zhengzhou (34°16′–34°58′ N; 112°42′–114°14′ E), the capital of Henan Province in Central China, covers an area of approximately 7446 km$^2$. The city is located in the south of the North China Plain and central Henan province in the north, limited by the Yellow River in the north and Mount Songshan in the west (Figure 1). As the largest city in central China, Zhengzhou has an estimated urban population of more than 15 million in 2020. The urban area of Zhengzhou is divided into four regions: Zhongyuan, Erqi, Huiji, Jinshui,

and Guancheng. Due to the continental monsoon climate in Zhengzhou [27], although the annual rainfall reaches 640.9 mm, 60% of the precipitation takes place from June to September during the summer, when there is an increased risk of urban flooding. For example, on 1 August 2019, a rainstorm hit Zhengzhou with rainfall exceeding 100 mm. As a result, some sections of roads in Zhengzhou were seriously flooded, compromising the regular traffic operation.

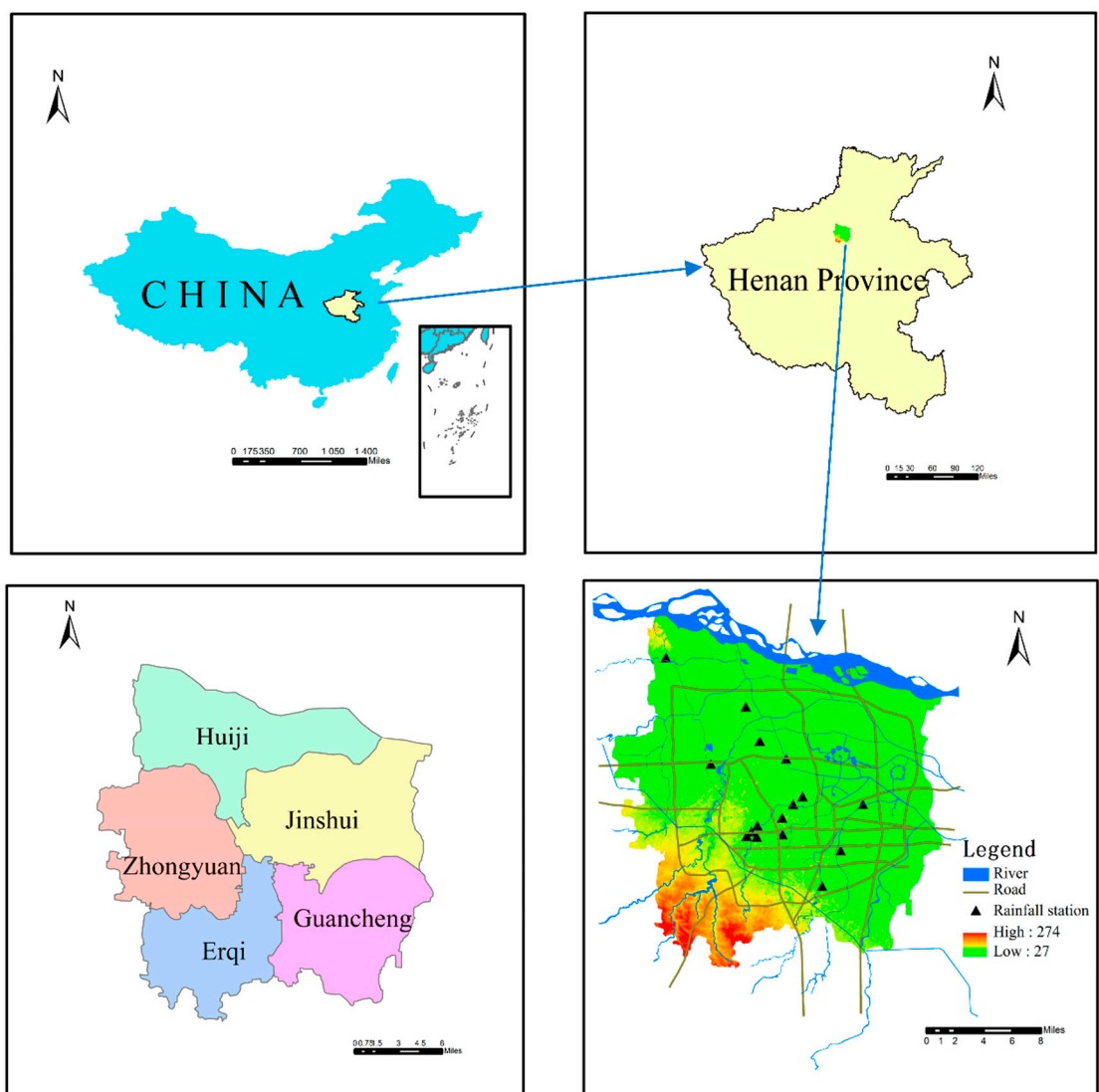

**Figure 1.** Location and partition of the study.

### 2.2. Date and Material

Some scholars have used statistical methods [28] or structured questionnaires [29] to analyze and study the historical floods, which have to do with identifying which factors have more weight on flood occurrence. But there are no universal guidelines for selecting flood conditioning factors in urban areas [30]. From the analysis of the hydrological process from rainfall to runoff, urban undersurface factors and hydrological meteorology factors can all affect the formation of a flood. Thus, three and eight factors were selected from the hydrometeorology and underlying surface, respectively, to conduct simulation research on urban flooding (Table 1).

**Table 1.** Descriptions of 11 impact factors and flood data in the study.

| Group | Factor | Abbreviation | Description |
|---|---|---|---|
| hydrometeorology factors | rainfall | R | the total amount of rainfall for a given rainfall |
| | rainfall duration | RD | the duration of a rainfall from the beginning to the end |
| | peak rainfall | PR | peak on the rainfall intensity curve |
| underlying surface factors | the proportion of roads | TPR | the ratio of road area to the overall regional area |
| | the proportion of woodlands | TPW | the ratio of the area planted with trees to the overall regional area |
| | the proportion of grasslands | TPG | the ratio of the area planted with grasslands to the overall regional area |
| | the proportion of water bodies | TPWB | the ratio of the area of lakes, rivers and other water bodies in the region to the overall regional area |
| | the proportion of building | TPB | the ratio of district building area to the overall regional area |
| | permeability | P | the water permeability of the land within the area |
| | catchment area | CA | the area of the subregion |
| | slope | S | regional mean topographic inclination |
| flood data | depth of the flood | DF | recorded value of the change in water level |

The rainfall data of flooded areas were obtained by using the Kriging method of space interpolation, the data of underlying surface factors were obtained by extracting from the 0.5 m high spatial resolution map of the Pleiades Satellite in May 2014, and the flooded data was obtained from the Zhengzhou Municipal Urban Management Bureau. The data about the locations and depths of the flooded area were extracted from the historical flooding records, which were collected from the monitoring equipment at each intersection. All hydrometeorological data were extracted from the Henan Meteorological Service.

Fourteen historical rainfall events from 2011 to 2017 with 10 min temporal resolutions were selected as sample data for the model. The rainfall events happened specifically on July 26 and August 3, 2011; July 2 and August 27, 2012; May 26 and August 15, 2013; June 9 and 19, 2014; May 26 and July 22, 2015; June 11, July 19 and 5 August, 2016; and 20 July, 2017. Each rainfall packet contained flood information from 3324 sample points. Drainage systems did not undergo any changes over the period the data was collected.

*2.3. Naive Bayes Algorithms*

Naive Bayes algorithm is one of the few classification algorithms based on probability theory of the classical machine learning algorithms. In some areas, its performance can be comparable to neural networks and decision-tree learning [31]. For the prediction task in this paper, it was necessary to calculate the probability that the prediction result belonged to according to the known correlation probability, and to take the depth category with the highest probability as the optimal output result.

In the training phase of the model, the input dataset was represented as $(x_1, x_2, x_3, \ldots, x_d, y_j)$. $X$ is the input vector, which contains the value of the hydrometeorology factors and underlying surface factors $(x_1, x_2, x_3, \ldots, x_d)$ in this study, and $y_j$ is the maximum depth of the flood in the $j$th sample point (Figure 2). $Y$ is the vector that is made up of $y_j$, where $j$ is the natural number from 1 to 3324. For the testing and prediction phase of the model, the input dataset was only $(x_1, x_2, x_3, \ldots, x_d)$. Then, for each input vector $X$, the value that can maximize the posterior probability $P(Y|X)$ is selected as its optimal output result. Based on Bayes' theorem, $P(Y|X)$ can be written as the formula:

$$P(Y|X) = \frac{P(Y)P(X|Y)}{P(X)} \tag{1}$$

where $P(Y|X)$ is the conditional probability of the sample input vector x relative to the output variable y, $P(Y)$ is the prior probability, and $P(X)$ is the attribute factor.

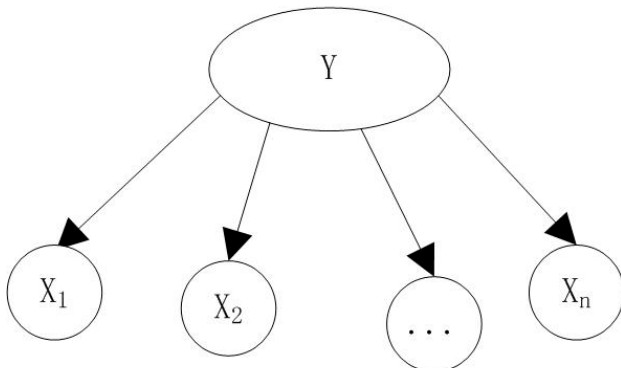

**Figure 2.** The structure of the Naïve Bayes.

For a given sample $X$, the attribute factor $P(X)$ is independent of the class tag, so $P(Y|X)$ is only related to $P(Y)$ and $P(X|Y)$. One of the assumptions of Naive Bayes is "attribute condition independence", which assumes that each network characteristic attribute independently has an attribute influence on the prediction results. Based on this assumption, the formula can be rewritten as:

$$P(Y|X) = \frac{P(Y)\,P(X|Y)}{P(X)} = \frac{P(Y)}{P(X)}\prod_{i=1}^{d} P(x_i|Y) \tag{2}$$

where, $d$ is the number of attribute factors, which in this article is 11 (i.e., the sum of hydrometeorology and underlying surface factors). $X_i$ is the value of $X$ on the $i$th feature.

$P(X)$ is equivalent to a constant for all the predictions. Based on the assumption of attribute independence, $NB$ can be expressed as:

$$h_{NB}(X) = \mathrm{argmax}P(Y)\prod_{i=1}^{d} P(x_i|Y) \tag{3}$$

Equation (3) means that the values of $P(Y)\prod_{i=1}^{d} P(x_i|Y)$ under different $Y$ values are obtained by inputting different vectors $X$, and $Y$, corresponding to the maximum $P(Y)\prod_{i=1}^{d} P(x_i|Y)$ value, is taken as the output quantity.

*2.4. Evaluation of Model Accuracy*

Model evaluation is an important step in the modeling and prediction process. Selecting reasonable model evaluation indexes can quantitatively analyze and evaluate the accuracy of model-based predictions. Standard evaluation indexes of prediction models mainly include the mean square error (MSE), root mean square error (*RMSE*), mean absolute error (MAE) and mean absolute percentage error (MAPE). The *RMSE* is derived from the square root of the MSE and is the same as the MSE and MAE. The larger the mean error is, the larger the value is. Therefore, *RMSE* is the most appropriate index to evaluate and analyze the prediction results of the model in the current study. Moreover, the qualified rate (*QR*) and the Nash–Sutcliffe efficiency coefficient (*NSE*) were also selected as the evaluation indexes of the model prediction accuracy. They are mathematically described as:

$$RMSE = \sqrt{\frac{1}{n}\sum_{i=1}^{n}(y_{si} - y_{oi})^2} \tag{4}$$

$$QR = \frac{N_{QSN}}{n} \times 100\% \tag{5}$$

$$NSE = 1 - \frac{\sum_{i=1}^{n}(y_{oi} - y_{si})^2}{\sum_{i=1}^{n}(y_{oi} - \overline{y}_o)^2} \tag{6}$$

where $y_{si}$ and $y_{oi}$ is the simulated value and the measured value of the flood at the point $i$, respectively; $N_{QSN}$ is the number of samples whose relative error < the allowable error, which was determined to be 20%; $\bar{y}_o$ is the average value of measured value, and n is the total number of samples. Lower values of *RMSE* and higher values of *QR* and *NSE* are used to indicate the performance superiority of the model.

## 3. Results and Discussions

### 3.1. Data Processing

The research object includes three hydrometeorological factors and eight under surface factors. The original data that quantitatively describe these factors have different attributes and sources, and there are a considerable number of them. Therefore, we need a thematic database to store and extract the data. A database focusing on urban flood forecasting was built by using Microsoft SQL Server Management Studio 10.2 to store a variety of data needed for the research in this study, which included extracting, transforming and loading data. The diversified data processed by the established database can be directly applied by the model.

### 3.2. Forecasting Model of Urban Waterlogging Flood Depth Based on Naïve Bayes Algorithm

From the established thematic data warehouse containing 14 historical rainfalls and corresponding flood depth data (2011–2017), the rainfall and water depth data on 9 June 2014; 5 August 2016; and 20 July 2017 were selected as test data set to evaluate the predictive performance of the model. Those other representative rainfall and water depth data represent most of the rainfall scenarios used to train the model. Each sample datum in both the training data set and the test data set contains three hydrometeorology factors (rainfall, rainfall duration, and peak rainfall), eight underlying surface factors (the proportion of roads, woodlands, grasslands, water bodies, building, permeability, catchment area, and slope), and one metric factor (depth of flooded areas).

Those dates were loaded into the SQL Server Data Tools, which is a supporting tool of the SQL Server used to build, debug, maintain, and refactor databases. After training the NB model with the training data set, we obtained the contribution of conditioning factors to the depth of flooding at most sample points. The result was R (1.00), PR (0.99), RD (0.96), CA (0.88), TPG (0.85), TPB (0.82), TPWB (0.75), P (0.62), TPR (0.10), TPW (0.08), S (0.03). The top eight conditioning factors are shown in Figure 3. Although the values of the contribution of TPR, TPW, and S were low, eleven related factors were still used to predict the depth of flooding with high accuracy. It can be seen from Figure 3 that the main causes of flooding are the hydrometeorology factors (rainfall, peak rainfall, and rainfall duration).

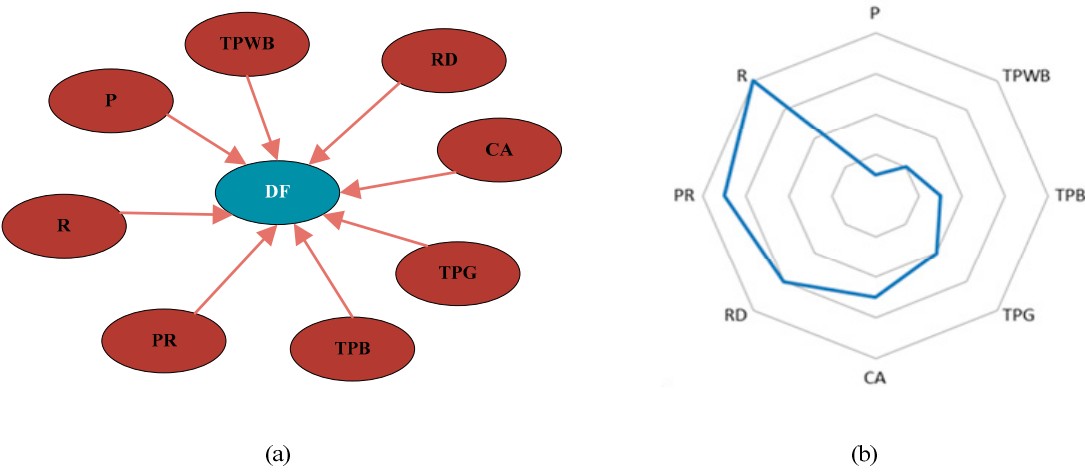

(a)                                                           (b)

**Figure 3.** Network diagram of dependencies: (**a**) Through training the NB model with training data, the top eight factors influencing the depth of urban flood were extracted from 11 predictive factors; (**b**) The radar map is used to display the ranking results of these eight factors. It can be seen that the contribution of rainfall to the formed flood depth is the largest and the proportion of water bodies is the smallest in the top eight factors.

Then, the test data set was input into the trained NB model; the predicted results of the test set are shown, taking 5 August 2016 as an example (Table 2).

**Table 2.** Simulation results of the test set by the NB model (5 August 2016).

| The Label Number of the Sample Points | Simulated Depth of Sample Points/m | Measured Value of Sample Points/m | Absolute Error/m | Relative Error/% |
|---|---|---|---|---|
| 1 | 0.27 | 0.28 | −0.01 | 3.5 |
| 2 | 0.13 | 0.132 | −0.002 | 1.5 |
| 3 | 0.24 | 0.27 | −0.03 | 11.1 |
| 4 | 0.13 | 0.132 | −0.002 | 1.5 |
| 5 | 0.139 | 0.142 | −0.003 | 2.1 |
| . . . | \ | \ | \ | \ |
| 3320 | 0.46 | 0.46 | 0 | 0 |
| 3321 | 0.37 | 0.36 | 0.01 | 2.7 |
| 3322 | 0.32 | 0.33 | −0.01 | 3.1 |
| 3323 | 0.28 | 0.259 | 0.021 | 8.1 |
| 3324 | 0.24 | 0.26 | −0.02 | 7.6 |

### 3.3. Evaluating the Performance of the Model

The *RMSE*, *QR*, and *NSE* between the predicted value and measured value were used to evaluate the accuracy of the model (Table 3). It can be seen from the results that the average *RMSE* of the three rainfall events was 0.062, the average *NSE* was 0.94, the minimum *QR* value was 83%, and the maximum value was close to 90%, indicating that the NB model is applicable to the prediction of flood depth. We entered the measured value and the model's predicted value of the model into the Microsoft graphics software Origin2017, and smoothed the image with a window of 200 points. Figure 4 shows the overall good consistency between the simulated value and the measured value, thus indicating the model's effectiveness in predicting the urban flood depth. However, it can also be seen that there is a large gap between the simulated value and the measured value in some local parts. For example, the simulated values from 2000 to 2500 and 2750 to 3000 are not as close to the measured values as the simulated values in other ranges. Their relative error values are still mostly within 20%, however, which is within acceptable limits. This may have something to do with the accuracy of the data collected.

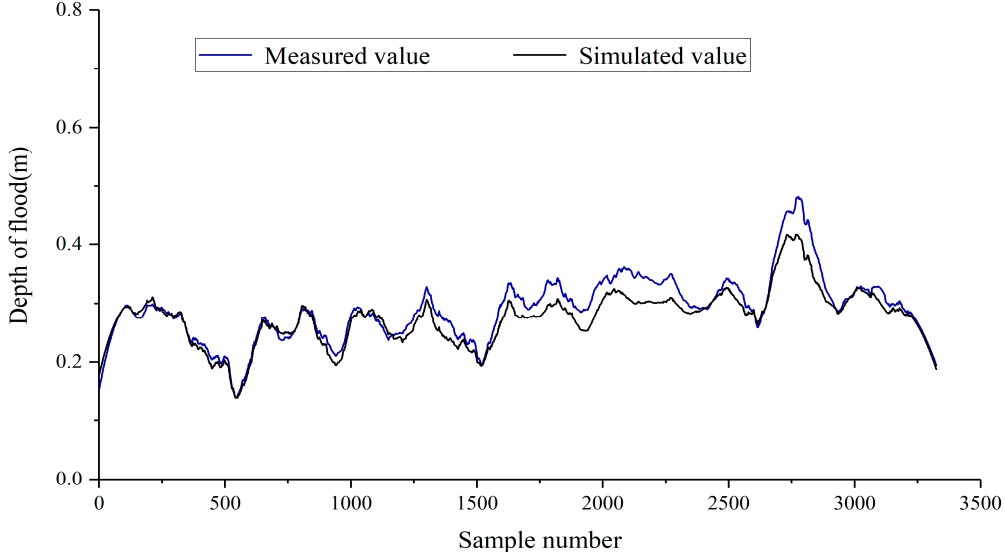

**Figure 4.** Fitting curves between simulated and measured values (5 August 2016).

**Table 3.** The value of the evaluation indexes.

| Index | *RMSE* (cm) | *QR* (%) | *NSE* |
|---|---|---|---|
| 9 June 2014 | 0.065 | 83 | 0.92 |
| 5 August 2016 | 0.059 | 89 | 0.97 |
| 20 July 2017 | 0.061 | 85 | 0.94 |
| Average value | 0.062 | 86 | 0.94 |

To compare the effectiveness of the Naive Bayesian prediction model, the information of the last rainfall was also loaded into SWMM to obtain the simulation results of the SWMM model. The SWMM model was modeled by the rest of the team and has been properly calibrated and tested, in which the pipeline overflow rate of the SWMM model is laid out as the water volume to the ground to get the flood depth as indicated in the literature [32]. The results are compared with the model in this paper (Table 4). The operation was carried out on a Dell laptop with an Intel(R) Core(TM) i5-6200U CPU @ 2.30 GHz 2.40 GHz.

**Table 4.** Comparison between the SWMM and NB models.

| Method | Time Consumed by the Model (Second) | *NSE* |
|---|---|---|
| SWMM | 65 | 0.82 |
| NB | 2 | 0.95 |

*3.4. Waterlogging Prediction under Different Rainfall Conditions in Zhengzhou City Based on Naive Bayesian Model*

When analyzing the causes of urban waterlogging, heavy rainfall events are often the dominant factors [33], which are also illustrated in Figure 3. In Zhengzhou, the rainfall pattern that produces urban waterlogging is usually a short duration heavy precipitation event. Therefore, we used an NB model to predict the depths of flooded areas caused by various levels of rainfall with different recurrence periods, which were once every 0.5, 1, 2, and 5 years. This study determined that the rainfall duration was 2 h, and the time step was 5 min (Figure 5, Table 5); the rainstorm intensity formula of the Zhengzhou urban area is as follows [34]:

$$i = \frac{262.0278 + 261.5298 \log T}{(t + 56.9709)^{1.3109}} \tag{7}$$

where $i$ is the intensity of rain, $T$ is the designed rainfall return period, and $t$ is the rainfall duration. The traditional Chicago rain pattern was used to determine the course of the rainfall to obtain peak rainfall and average rainfall [35]. The Chicago rainfall process line model uses the precipitation peak coefficient r (0 < r < 1), which is defined as the ratio of the time before the peak intensity to the total duration to describe the peak time of the rainstorm. The precipitation peak coefficient r = 0.4 divides the rainstorm into two parts—the pre-peak and the post-peak.

**Table 5.** Rainfall data under different design rainfall return periods.

| Return Period (Year) | Rainfall (mm) | Peak Rainfall (mm/h) | Rainfall Duration (min) |
|---|---|---|---|
| 0.5 | 27 | 54 | 120 |
| 1 | 38 | 78 | 120 |
| 2 | 51 | 102 | 120 |
| 5 | 66 | 133 | 120 |

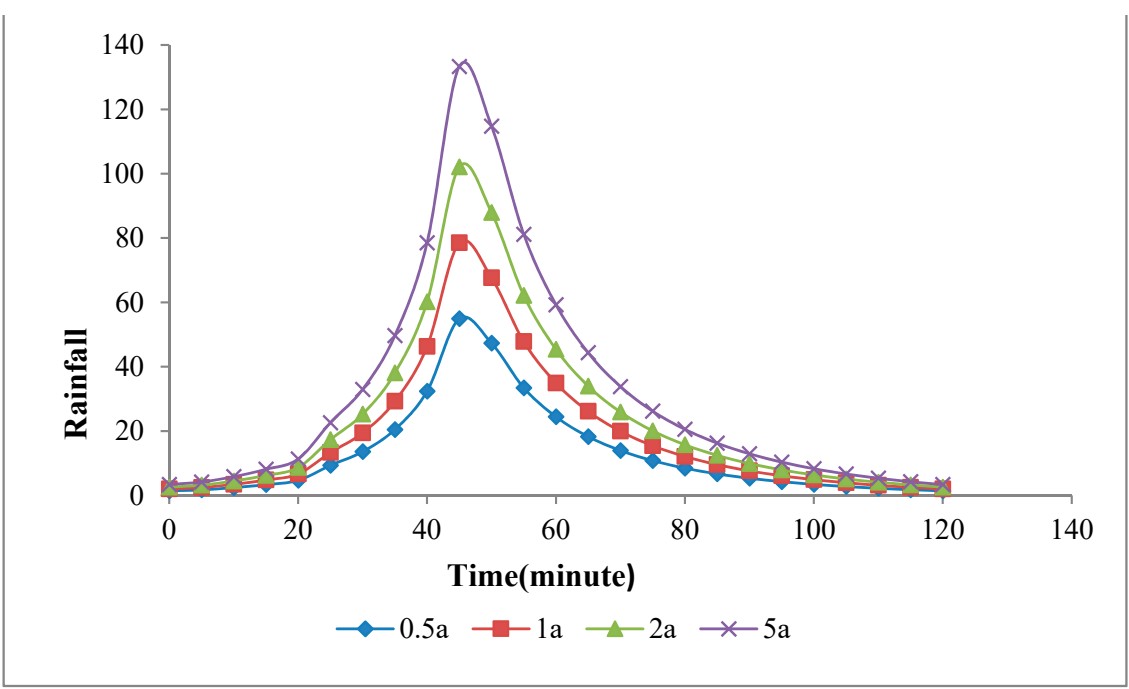

**Figure 5.** Designed rainstorm process.

The hazard of urban waterlogging is divided into four level classes according to the depth of flooded urban areas: 0–3, 3–10, 10–25, and >25 cm. The flooded area information of Zhengzhou City was obtained by adding the rainfall data under the different rainfall conditions (four different rainfall return periods) into the NB model, which was then used to produce the urban inundation map through ArcGIS. The urban inundation map includes the location, depth, and range of waterlogging (Figure 6). According to the picture information, we can clearly see which parts of the city are more prone to flooding, so as to effectively deal with the emergency of a flood or to rectify and deal with these areas in advance.

According to the results (Figures 6 and 7), with an increase in the rainfall return period, Zhengzhou city will experience an increasing trend in the number, depth, and surface area of the water accumulation area. In the scenario of the semi-annual rainfall recurrence period, water accumulation is predicted to occur in 47 regions, among which 36 were in the 3–10 cm class, 8 were in the 10–25 cm class and no region had more than 25 cm of flood. In the once-a-year rainfall return period, the number of estimated flooded areas increased to 131 (100 at 3–10 cm, and 29 at 10–25 cm). In the once-every-two-years rainfall return period, the number of estimated flooded areas reached 366. The number of submerged areas >25 cm was predicted to be 12, a value six times higher than during the same level of the once-a-year rainfall return period. Under the once-every-five-years rainfall return period, the number of flooded areas was more than 540, (48.6% higher than that of the once-every-two-years rainfall return period). The contribution of a 3–10 cm depth was 55.3% and the depth of >25 cm was 23, which was 4%. The proportion of the areas whose depth >25 cm in the flooded areas is still rising.

Based on the urban inundation map (Figure 6), it was found that the probability of an inundation depth of the sample points >3 cm in the old urban area of Zhengzhou, such as the Jinshui district and Guancheng District, is higher than that of other districts, which suggests that these areas are more prone to urban flooding. Through field investigation and analysis of map building data, we found that the places prone to water accumulation are mainly located under overpasses, old roads, sections along rivers and some construction sections in these old urban areas. The main reasons for water accumulation in these areas are as follows [36]:

In construction areas, construction hinders the discharge of water.

The design standard of the rainwater pipe network is low, which cannot meet the current demand. The old urban district of Zhengzhou had its pipe network laid earlier, the old design standard is lower, the pipe diameter is smaller, and the number of pipes in the network is fewer.

The rainwater network has silted up, resulting in poor drainage. After the rainwater network was laid, the maintenance was not in place. With the increase of the service life of the drainage system, part of the sediment silts the bottom of the pipe, leading to poor drainage. When it rains, ground rainwater cannot drain into the rainwater pipe in time, accumulating water.

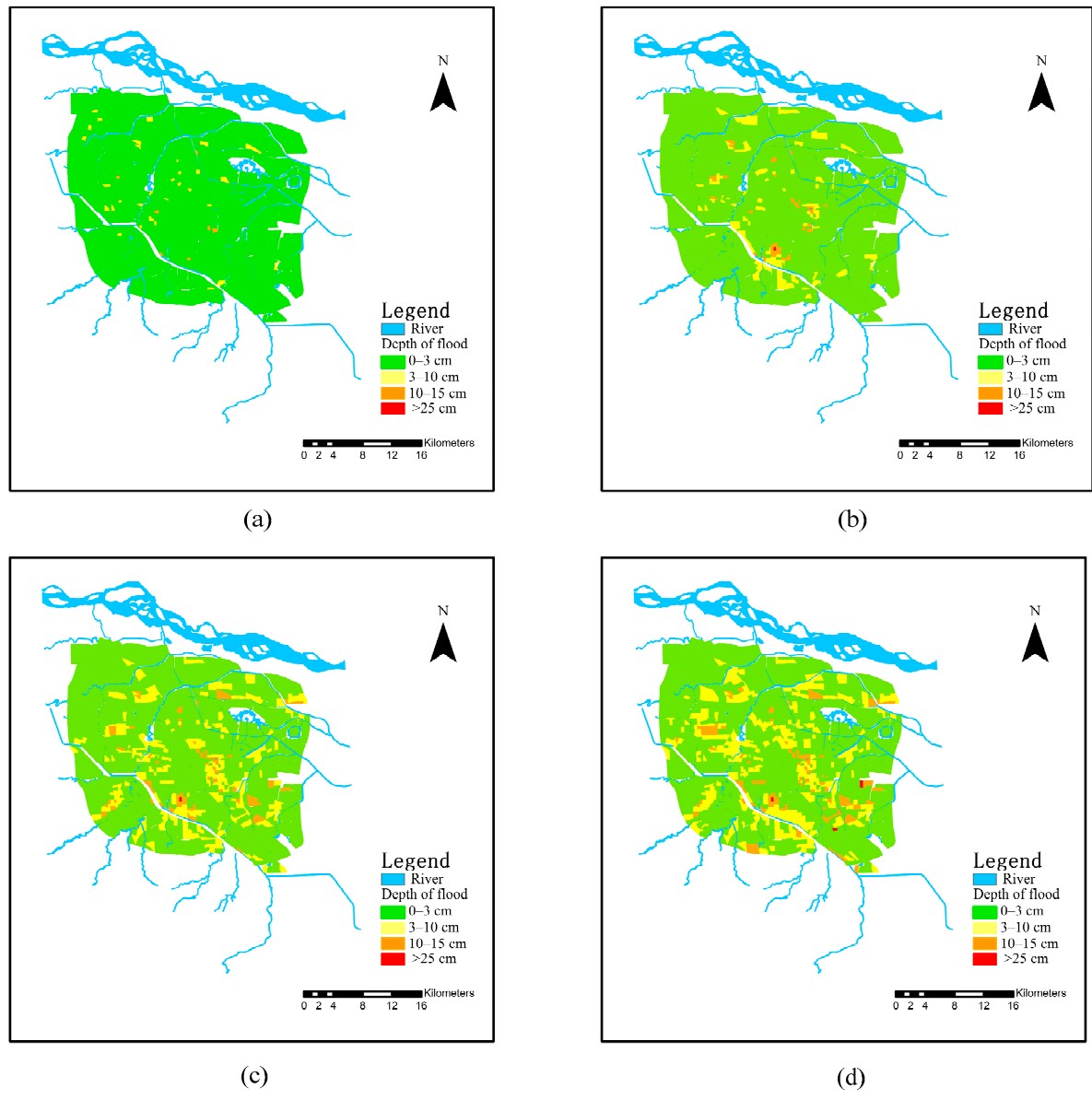

(a)

(b)

(c)

(d)

**Figure 6.** The urban inundation map under different designs of rainfall return periods: (**a**) 0.5a, (**b**) 1a, (**c**) 2a, (**d**) 5a.

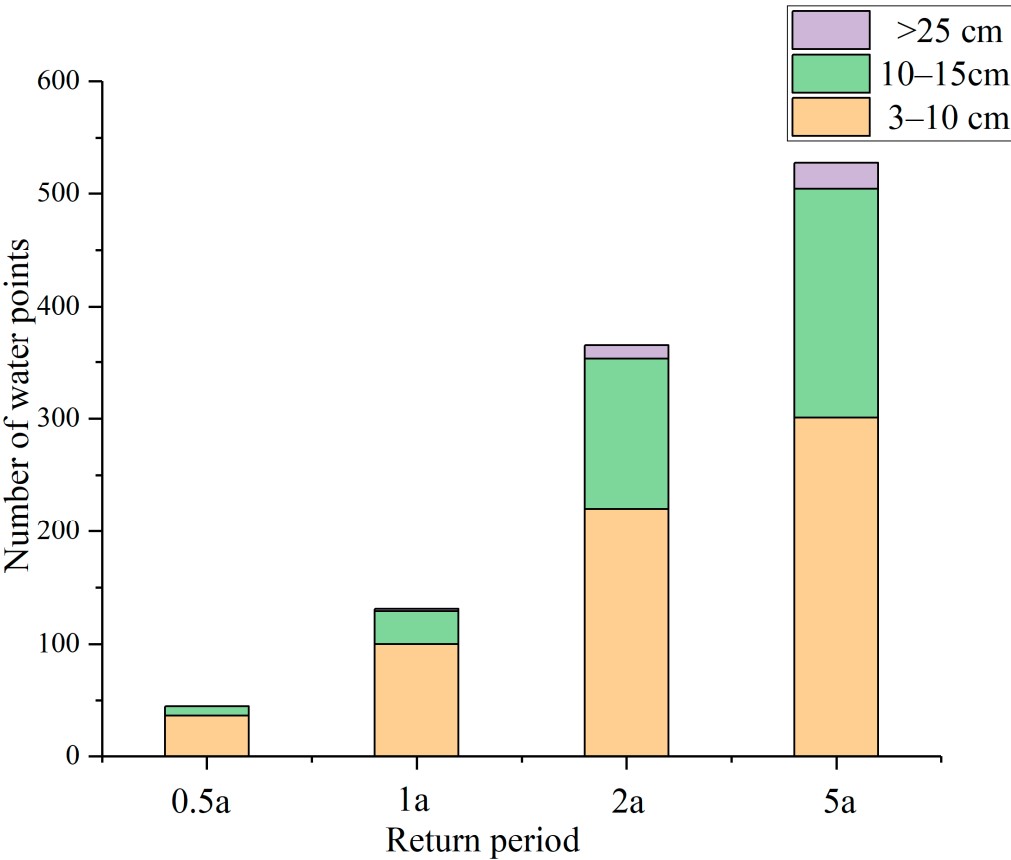

**Figure 7.** Inundation infographic under different return periods.

## 4. Conclusions

In this study, the Naïve Bayes theory is applied to the depth prediction of urban flooding. The *RMSE*, *QR*, and *NSE* were used to evaluate the performance of the NB predictive model. The results show that the NB model predicted urban flood depth with good performance. The data of flood depth predicted by the NB model based on different rainfall scenarios were imported into ArcGIS to generate urban inundation map. This revealed that the flooded conditions were more severe after the rainfall return period of once-two-year in Zhengzhou City, and the flooded areas mainly occurred in the old urban areas of the city and the main reasons were given. It is an effective way to reduce the possibility of urban flood in Zhengzhou to clean up the sediment of drainage system and increase drainage equipment regularly. Meanwhile, based on the geographic information about flood inundation obtained by ArcGIS, detailed information (including depth and location) on the flood caused by this rainfall can be provided to the majority of urban residents, so as to give early warning and allow the city's flood control work to have enough time to prevent and make decisions based on the damage predicted to be caused by the flood in advance.

**Author Contributions:** H.W. (Huiliang Wang): writing—original draft preparation, methodology and investigation. H.W. (Hongfa Wang): formal analysis, resources and validation. Z.W.: conceptualization, resources and supervision. Y.Z.: software and data curation. All authors have read and agreed to the published version of the manuscript.

**Funding:** The study was funded by the Key Project of National Natural Science Foundation of China (No: 51739009), Natural Science Foundation of China (51879242), Science and Technology Innovation Talents Project of Henan Education Department of China (21HASTIT011), Young backbone Teachers Training Fund of Henan Education Department of China (2020GGJS005), and Excellent Youth Fund



of Henan Province of China (212300410088). The authors thank the anonymous reviewers for their valuable comments.

**Conflicts of Interest:** The authors declare that they have no known competing financial interests or personal relationships that could have appeared to influence the work reported in this paper.

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
