# Peer review of "Using Multi-Factor Analysis to Predict Urban Flood Depth Based on Naive Bayes"

_water, doi:10.3390/w13040432_

Round 1

Reviewer 1 Report

The authors have made a good job in introduction, clearly stating where scientific findings stand right now, and what is their objective/aim. The introduction is mostly well written. Their methods can be improved but in general they are of good quality. However, there are several issues that need to be improved

Major comments:

The data described in Table 1 are not adequately described. I remind the authors that the data and methodology should be described in a way that can be replicated, so they should include all the relevant details. For example the authors should provide more information on what “permeability”  data layer is as well as the other factors mentioned in Table 1.

The authors have done a poor job in explaining the added value and the practical implications of their application in conclusions section or previously. The stay to the presentation of their results

The authors do not demonstrate adequately the validation of their method. The study needs a significant improvement (also in a graphical way) .

Minor comments:

1) The authors miss a field of flash flood prediction in urban areas that has to do with identifying which factors weight more in flood occurrence. Previous studies have connected statistically some these factors in spatial terms. I suggest including some comment about this field (please see references).

Diakakis, M., Priskos, G., & Skordoulis, M. (2018). Public perception of flood risk in flash flood prone areas of Eastern Mediterranean: The case of Attica Region in Greece. International journal of disaster risk reduction28, 404-413.

Faccini, F., Luino, F., Paliaga, G., Sacchini, A., Turconi, L., & de Jong, C. (2018). Role of rainfall intensity and urban sprawl in the 2014 flash flood in Genoa City, Bisagno catchment (Liguria, Italy). Applied Geography98, 224-241.

2) Figure 1 needs improvement. The authors should include scale in all maps and make sure they keep the same style in all four images. For example the upper right image does not have a proper framework like the rest. In addition, it would be ideal to keep the fonts that the journal recommends and keep the same size of fonts.

3) Please revise the caption of Table 1. Look for grammatical errors.

4) Figure 3 text is not readable. The fonts are pretty small. Please enlarge the fonts and keep the journal’s font and other standards.

5) the last part of section 3.3 is more related to the methodology rather than the results. I believe it should be moved to methodology.

6) what do the authors mean by “Scope of water accumulation”. Please clarify for the readers.

7) The authors should generally acknowledge in their study that floodwaters follow  hydrodynamic laws expressed by hydraulics-related equations and are affected by obstructions to flow (even minor ones) that present very significant and localized fluctuations . For example we have seen many times in urban floods individuals struggling to cope with running water or drive through floodwaters and 10-20m away from this area others standing outside the inundated area. In a few meters distance we see water depth going from 0cm to 1 or 1.5m. So the authors have to stress that their model is based on assumptions that lead to a simplified map of water depth. Also the authors should explain why near the drainage network there is no depth data on their map or in most cases lower depth. Normally, from waters overflowing from the drainage network we see higher or much higher floodwater depth.

Reviewer 2 Report

There are several typos and grammatical errors that largely make the organization of the manuscript difficult to follow. The authors should check and correct them.

Author Response

Dear Editors and Reviewers:

Thank you very much for your letter and for the reviewers’ comments concerning our manuscript entitled “Using multi-factor to predict urban flood depth based on Naive Bayes” (water-1047323). Those comments are all valuable and very helpful for revising and improving our paper, as well as the important guiding significance to our research. We have studied comments carefully and have revised the article, which were marked in yellow bold.

 Comment:There are several typos and grammatical errors that largely make the organization of the manuscript difficult to follow. The authors should check and correct them.

Response to comment: Thank you very much for your careful reading of our manuscript. Author is not a native English speaker. After receiving your suggestions, we have revised and improved the content and wording of the article. For example, line 51, 64, 91and so on. I hope this modification can meet your satisfaction.

Finally, special thanks to you for your good comments again. Because of your valuable suggestions, we found some shortcomings in our original manuscript. Through your comments, we identified the shortcomings of our original paper and further perfected our research. We will improve the abilities of scientific research and make more achievements according to your suggestions in the future work. And we sincerely hope that we can learn more from you.

Best regards.

Yours sincerely,

Hongfa Wang

Reviewer 3 Report

This paper presents a novel application of the Naive Bays model to the simulation of flood hazard. However, the paper must be improved to be publishable in water. Here are my major comments:

1 - The introduction is fine. On line 47, I would add other references concerning hydrological and hydrodinamic models, including the following:

Galuppini et al. (2020). A unified framework for the assessment of multiple source urban flash flood hazard: the case study of Monza, Italy. Urban Water Journal, 17, Issue(1), 65-77.
2 - On line 63 and in other parts of the work, the Authors speak about scope of flood. What does this mean? I would use another term.

3 - line 81-82. The Authors ought to say that under changing urban drainage systems only physically-based models would be successful.

4 - line 121 starts abruptly. Is seems that the previous part of the paper has been wrongly deleted.

5 - Equation 2. What is hrb? What is arg max?

6 - subsection 2.3. Plese explain what is the final output of the model? Is it the flood depth in the generic grid element?! Or is it an average value in the territory?

7 - Please improve the quality of Figure 3.

8 - section 3.2. The results should be distinguished between training and testing!

9 - lines 169-173. It's unclear how this relates to the methodology of NB?

10 - Table 2. What does number mean?

11 - Figure 4. What does deep of flood mean? What does sample mean?

12 - Table 4. Here swmm appears abruptly? Do you have the SWMM model of the area? SWMM alone doesn't enable modelling flood propagation. It should be coupled with FLO2D, as shown by Galuppini et al. (2020). Was the SWMM model suitably calibrated?

13 - lines 208-213 and whole section 3.4. How are you sure that the model is suitably trained to be applied for this?

Reviewer 4 Report

In my opinion, the manuscript presents an interesting study for the calculation of flooding in extensive regions, and from general variables of the terrain and climate. Therefore, the study has options to be replicated in different locations. These characteristics make it suitable for application in areas where the availability of information is limited.

However, I believe that the manuscript has several aspects to be improved. The two main aspects would be, on the one hand, the quality of some of the figures presented. And on the other hand, and much more important, I consider that the manuscript in its present form suffers from a lack of explanation of the results obtained during the "training" phase of the model. Without further explanation, the results presented in the manuscript about the "Test" phase of the model and its validation process may generate multiple doubts and uncertainties.

Therefore, in addition to the minor comments that the authors can observe in the attached document, I believe that the authors should make an important effort to improve the presentation of the results, relationships, weights of variables, selection of principal variables, and formulations obtained during the "training" phase of the model. In this way, and in view of these results, the confidence in the results shown relative to the "Test" phase can be considered reliable.

On the other hand, and although it appears as a minor commentary in the attached document, I believe that the authors should analyze the possible spatial patterns in the results obtained, since a clear difference was appreciated in the level of adjustment of the measured and predicted values according to the number of control points considered.

Round 2

Reviewer 1 Report

The authors addressed the issues in a satisfying way. I recommend considering the study for publication in its current form.

Author Response

Thank you very much for your suggestions on perfecting my manuscript.

Reviewer 3 Report

For some previous comments of mine, the Authors did not reveal the part of the paper in which they intervened to reply. I would like to find the modifications of the paper in reply to these comments:

  1. Equation 2. What is hrb? What is arg max? Please, intervene in the paper.
  2. subsection 2.3. Plese explain what is the final output of the model? Is it the flood depth in the generic grid element?! Or is it an average value in the territory? Please intervene in the paper
  3. Table 2. What does number mean? Please explain in the table, e.g. in the column heading 
  4. Table 4. Here swmm appears abruptly? Do you have the SWMM model of the area? SWMM alone doesn't enable modelling flood propagation. It should be coupled with FLO2D, as shown by Galuppini et al. (2020). Was the SWMM model suitably calibrated? Please, explain in the text that a previously calibrated SWMM model was available

Reviewer 4 Report

Dear Authors,

The changes made to the manuscript have solved a large part of the comments made. However, in my opinion, the main criticism of the manuscript has not been adequately resolved. It is logical, I believe, because I sent the revision of the manuscript on January 4, and on January 5 I received communication that the authors had already sent the corrections to the manuscript. In a single day, I do not think it is very likely that it will be possible to correct and improve the quality of the manuscript.

Therefore, I still consider that the "training" phase of the model is not well described, and leaves many doubts about it.

It is indicated that 11 variables have been used in the study, and yet Figure 3 shows only 8 variables. What happened to the other three variables? Is the weight of each of the 11 (or eight) variables homogeneous throughout the study area? What is the weight of each of the variables in the assignment of a flow depth value?

I believe that in this part of the manuscript there continues to be much uncertainty about how the authors have solved the "training" phase of the model.

Some comments have been added to the manuscript in the attached document. But since this document already had parts of the text highlighted, I indicate below the position (lines) and commentary made:

- Lines 180-182:
It is this radar map representative for all locations of the study area?

If not, you should describe variations along study area.

TPR; TPW; and S factors were finally not used? Why?

- Lines 195-196:
But the Figure 3 shows only 8 factors, not 11. Why?

Furtheremore, if you are weighting each factor from the "training" phase, what is the factor for each factor?

- Line 241:
You are not making a risk assessment. You are making a hazard assessment, because you are not considering exposed elements and its flood vulnerability.

Please change the word to be consistent with the study you have conducted.

Round 3

Reviewer 3 Report

The Authors have replied to my comments. I only recommend that the paper be proofread.

Reviewer 4 Report

Dear Authors,

In my opinion the manuscript has been satisfactorily improved, and ready for publication. The only aspect that I think should be improved is the English language. You should review it.